# HbA1c and Aortic Calcification Index as Noninvasive Predictors of Pre-Existing Histopathological Damages in Living Donor Kidney Transplantation

**DOI:** 10.3390/jcm9103266

**Published:** 2020-10-12

**Authors:** Kosuke Tanaka, Shigeyoshi Yamanaga, Yuji Hidaka, Sho Nishida, Kohei Kinoshita, Akari Kaba, Toshinori Ishizuka, Satoshi Hamanoue, Kenji Okumura, Chiaki Kawabata, Mariko Toyoda, Asami Takeda, Akira Miyata, Masayuki Kashima, Hiroshi Yokomizo

**Affiliations:** 1Department of Surgery, Japanese Red Cross Kumamoto Hospital, Kumamoto 861-8520, Japan; kosuket0814@gmail.com (K.T.); yuji.h65@gmail.com (Y.H.); s.nishida06@gmail.com (S.N.); kouhei7687@gmail.com (K.K.); momiji.akanashi@gmail.com (A.K.); kenjiokumura@kyudai.jp (K.O.); h-yokomizo@kumamoto-med.jrc.or.jp (H.Y.); 2Department of Nephrology, Japanese Red Cross Kumamoto Hospital, Kumamoto 861-8520, Japan; m03002ti@jichi.ac.jp (T.I.); ririgugu@yahoo.co.jp (S.H.); chiaki-kawabata@hotmail.co.jp (C.K.); tmariko0413@gmail.com (M.T.); drmiyata19561@yahoo.co.jp (A.M.); m-kashima@kumamoto-med.jrc.or.jp (M.K.); 3Department of Nephrology, Japanese Red Cross Nagoya Daini Hospital, Aichi 466-8650, Japan; asamit@nagoya2.jrc.or.jp

**Keywords:** renal aging, renal function recovery, HbA1c, aortic calcification index

## Abstract

We previously reported that allografts from living donors may have pre-existing histopathological damages, defined as the combination of interstitial fibrosis (ci), tubular atrophy (ct), and arteriolar hyalinosis (ah) scores of ≧1, according to the Banff classification. We examined preoperative characteristics to identify whether the degree of these damages was related to metabolic syndrome-related factors of donors. We conducted a single-center cross-sectional analysis including 183 living kidney donors. Donors were divided into two groups: chronic change (ci + ct ≧ 1 ∩ ah ≧ 1, *n* = 27) and control (*n* = 156). Preoperative characteristics, including age, sex, blood pressure, hemoglobin A1c (HbA1c), aortic calcification index (ACI), and psoas muscle index (PMI), were analyzed. Comparing the groups, the baseline estimated glomerular filtration rate was not significantly different; however, we observed a significant difference for ACI (*p* = 0.009). HbA1c (*p* = 0.016) and ACI (*p* = 0.006) were independent risk factors to predict pre-existing histopathological damages, whereas PMI was not. HbA1c correlated with ct scores (*p* = 0.035), and ACI correlated with ci (*p* = 0.005), ct (*p* = 0.021), and ah (*p* = 0.017). HbA1c and ACI may serve as preoperative markers for identifying pre-existing damages on the kidneys of living donors.

## 1. Introduction

Renal transplantation is the best option for patients with end-stage renal disease (ESRD) [1]. In Japan, the scarcity of deceased donors demands the need for new marginal living donors [1]. As the long-term ESRD risk for living donors is higher than that for the general health population [2,3], marginal living donors should be carefully selected [4,5,6].

We have recently reported that healthy living donors may have pre-existing histopathological damages at baseline biopsy (1 h after the reperfusion) [7]. This finding of chronic change (CC), defined by the combination of interstitial fibrosis (ci), tubular atrophy (ct), and arteriolar hyalinosis (ah) scores according to the Banff classification [8] (ci + ct ≧ 1 ∩ ah ≧ 1), is strongly associated with suboptimal recovery of the renal function in living donors 1 year after donation [7]. This combination represents the extent of renal chronic deterioration. Positive ci and ct scores correspond to the interstitial fibrosis/tubular atrophy (IF/TA), which is a final pathway and prognostic factor of chronic kidney damages [9,10]. A positive ah score corresponds to the chronic afferent arteriolar change observed in metabolic syndromes, including hypertension and diabetes [11,12]. Furthermore, the impact of the chronicity score was independent of the actual age [7].

This age-histology discrepancy implicated the fact that even eligible, healthy living donors with subclinical metabolic syndrome could have pre-existing histopathological damages. Many clinical studies suggested that metabolic syndromes such as hypertension [13,14,15], hyperglycemia [16,17], and dyslipidemia [18,19] were correlated with the deterioration of renal function in the general population, and subclinical signs of aging, a consequence of metabolic syndrome, such as aortic calcification and sarcopenia were common in patients with chronic kidney disease (CKD) [20,21]. However, the relationship between these subtle signs of metabolic syndromes and pre-existing histopathological damages in the healthy living donors is unclear.

Metabolic syndromes are also associated with glomerular hyperfiltration [22]. Increased glomerular pressure and hypertrophy by altered hemodynamics induces glomerular hyperfiltration and finally results in renal deterioration [23]. Remnant kidneys are also injured by donation-induced glomerular hyperfiltration after living donation [24]. Therefore, the subtle signs of metabolic syndromes are more critical in living donors.

Furthermore, noninvasive clinical markers would be preferred to invasive biopsies for living donors, because we could then preoperatively forecast histopathological damages and factor them into the living donor selection process.

Thus, we aimed to identify the predictive metabolic syndrome-related factors that would forecast pre-existing histopathological damages on the kidneys of living donors.

## 2. Materials and Methods

### 2.1. Study Population

This was a single-center cross-sectional study. Between May 2011 and Jan 2020, a total of 187 living kidney transplantations were performed at the Japanese Red Cross Kumamoto Hospital. Four cases were excluded due to unavailability of data on baseline biopsies. The 183 remaining cases were divided into two groups, according to our previous report [7] as previously noted: CC group (ci + ct ≧ 1 ∩ ah ≧ 1, *n* = 27) and control group (*n* = 156). We analyzed preoperative characteristics as possible predictive factors, including age, sex, tobacco use, blood pressure, hypertension, HbA1c concentration, uric acid concentration, low-density lipoprotein (LDL) concentration, ACI, body mass index (BMI), 3rd lumber PMI, and preoperative estimated glomerular filtration rate (eGFR). HbA1c, ACI, LDL, and PMI were examined as clinical indicators of hyperglycemia, aortic calcification, dyslipidemia, and sarcopenia, respectively. The detailed definitions and measuring methods of ACI and PMI are described in the subsequent chapters in this manuscript.

All donors were selected by strictly complying to the Japanese donor selection criteria [25]. This study was approved by the Institutional Review Board of Japanese Red Cross Kumamoto Hospital (study approval number 411). The review board waived the requirements of informed consent according to the nature of this research. None of the transplant donors were from a vulnerable population, and all donors or their next of kin provided freely given written informed consent.

### 2.2. Pathological Diagnosis

Baseline kidney biopsy was defined as a biopsy performed at 1 hour after reperfusion during the kidney transplantation [26]. Baseline biopsy data were collected retrospectively from the pathological reports. No other biopsies with different timings or causes (e.g., one-year protocol biopsy or episode biopsy) were included in this study. A Banff classification was used to assess pre-existing histopathological damages [8]. This classification enabled us to generalize and compare the degree of renal damages for the further analysis on living donor renal transplant recipients.

According to the Banff classification [8], histopathological findings were classified as ci, ct, and ah; ci referred to interstitial fibrosis, ct to tubular atrophy, and ah to arteriolar hyalinosis. Based on the cortical area percentage, the ci was classified as minimal (≦5%), mild (6–25%), moderate (26–50%), or severe (≧50%), which corresponded to ci-0, ci-1, ci-2, and ci-3, respectively. Ct was similarly categorized as ct-0, ct-1, ct-2, and ct-3. Ah was classified as none, mild-to-moderate, moderate-to-severe, or severe, corresponding to Banff scores of ah-0, ah-1, ah-2, and ah-3, respectively. As previously reported [7], we defined pre-existing histopathological damages as the combination of ci, ct and ah scores (ci + ct ≧ 1 ∩ ah ≧ 1).

### 2.3. Recording and Assessment of Clinical Data

We retrospectively collected all clinical data from medical records. The blood pressure in most donor cases was determined from 24-h blood pressure monitoring, and the average values of the systolic blood pressure obtained in the afternoon were adopted. We defined hypertensive donors as those who had an average systolic blood pressure of ≧ 140 mmHg, or those who took ≧1 antihypertensive drugs. HbA1c data were collected from the initial drawing samples (mainly three months before donor nephrectomy). In 21 cases, HbA1c measured using the Japan Diabetes Society (JDS) methods were converted into the National Glycohemoglobin Standardization Program (NGSP) HbA1c value according to the following formula [27]: NGSP value (%) = 1.02 × JDS value (%) + 0.25%. HbA1c was also described in accordance with the International Federation of Clinical Chemistry and Laboratory Medicine (IFCC) and reported as mmol/ml according to the following formula [28]: IFCC value (mmol/mol) = 10.93 × NGSP value (%)–23.50 mmol/l. eGFR was calculated by the formula recommended by the Japanese society of nephrology as [29]: eGFR (ml/min/1.73 m^2^) = 194 × Cr^1.094^ × Age^0.28^ in male donors and 194 × Cr^1.094^ × Age^0.28^ × 0.739 in female donors.

### 2.4. Aortic Calcification Index

ACI was examined as a clinical indicator of aortic calcification. ACI represents the calcification proportion of the abdominal aorta [30] and is calculated as the sum of the aortic calcifications evaluated from multidetector computed tomography (CT) images, with a window level of 30 Hounsfield unit (HU) and window width of 260 HU. We did not set the cutoff levels for the ACI analysis because the contrast between calcification and a healthy aortic wall was obvious enough not to rely on an arbitrary HU cutoff. ACI (%) was calculated by the following formula: ACI  =  (total score for calcification on all 10 slices)/120 × 100 (%). The score of calcifications was manually assessed by dividing the aorta into 12 sectors on each slice. Ten slices above the bifurcation of the aorta were added at every 10 mm interval. The ACI of the diabetic hemodialysis patients has been reported to be 57.3 ± 22.1%, and the ACI of predialysis chronic kidney disease patients has been reported to range from 0% to 76.6%, with a median of 11.4% [30,31].

### 2.5. Third Lumber Psoas Muscle Index

PMI is one of the methods used to evaluate the degree of sarcopenia [32]. The total areas of the right and left psoas muscles at the L3 level were measured by a manual tracing method using preoperative CT imaging. PMI was calculated as the division of the total psoas muscle area by muscle area square height (cm^2^/m^2^). According to the prior study for living liver donors, the normal value of PMI was 8.85 ± 1.61 cm^2^/m^2^ (mean, standard deviation) in men and 5.77 ± 1.21 cm^2^/m^2^ in women [32].

### 2.6. Statistical Analysis

All data were analyzed using IBM^®^ SPSS^®^ Statistics, version 25 (IBM Corp., Chicago, IL, USA). Data were expressed as the median and interquartile ranges for continuous data. In addition, for continuous data, a student’s t test, Mann–Whitney test and Kruskal Wallis tests were also used depending on the distribution of the data. The Chi-squared (χ^2^) test was used for categorical data. Preoperative factors were analyzed with the univariable and multivariable logistic regression analyses after controlling them simultaneously for potential confounders. The interaction analysis was used to identify the relationship between male and female donors. The variables included age, sex, tobacco use, blood pressure, hypertension, HbA1c, uric acid, LDL, ACI, BMI, PMI, and eGFR. The multivariable analysis was performed using four potential predictors: tobacco use, HbA1c, uric acid, and ACI, because their *p*-values were less than 0.1. The optimal cutoff points of preoperative continuous data in the prediction of chronic kidney damages were determined using a ROC curve analysis, and the AUC was calculated to assess the accuracy. A *p*-value < 0.05 was considered statistically significant. The authors have followed the suggestions of the Strengthening the Reporting of Observational Studies in Epidemiology statement guidelines for reporting observational studies [33].

## 3. Results

### 3.1. Donor Characteristics

The donor characteristics are shown in Table 1. The median age of all donors was 58 years. Sixty-five donors (35.3%) were male. Twenty donors (10.9%) were older than 70 years of age. Donors with a history of tobacco use and hypertension were 27.2% and 32.1%, respectively. The psoas muscle index (PMI) was 5.88 (5.05–7.21) cm^2^/m^2^ in male donors and 3.90 (3.41–4.42) cm^2^/m^2^ in female donors.

There was a significant difference between the chronic change (CC) group and control group regarding tobacco use (*p* = 0.035), uric acid (CC vs. control: 5.4 mg/dL (4.6–6.3) vs. 4.8 mg/dL (4.1–5.6), *p* = 0.029), and ACI (CC vs. control: 3.33 (0.0–13.3) vs. 0.83 (0.0–4.2), *p* = 0.009). The age and preoperative estimated glomerular filtration rate (eGFR) did not differ between groups. In the CC group, 18 cases (66.7%) had minimal ci (ci-0), whereas there was only one case (3.7%) with minimal ct (ct-0). No cases had minimal ah (ah0). All CC group donors had identical ci and ct (ci + ct ≧ 1). Living donors who were older than 70 years showed a tendency for positive pathological scores: ci1–3 (25%), ct1–3(50%), and ah1–3 (45%).

The frequency of ACI is demonstrated in Figure 1. The ACI was not normally distributed and 0% in 84 cases (45.9%). Eight donors (29.6%) in the CC group and 76 donors (48.7%) in the control group had an ACI of 0%. Eighteen donors (66.7%) in the CC group and 137 donors (87.8%) in the control group had an ACI of less than 10%.

### 3.2. Preoperative Predictors

Table 2 shows the results of the logistic regression analysis. The univariable analysis demonstrated that there was a significant difference in hemoglobin A1c (HbA1c) (odds ratio [OR] = 1.144 (1.023–1.278), per 0.1%, per 1.1 mmol/mol, *p* = 0.018), uric acid (OR = 1.369 (1.011–1.853), per 1 mg/dl, *p* = 0.042), and ACI (OR = 1.413 (1.132–1.763), per 5%, *p* = 0.002). HbA1c (adjusted OR [aOR] = 1.190 (1.032–1.371), per 0.1%, 1.1 mmol/mol, *p* = 0.016) and ACI (aOR = 1.445 (1.110–1.881), per 5%, *p* = 0.006) were revealed as independent risk factors for pre-existing histopathological damages at the baseline biopsy. HbA1c and ACI remained as robust independent factors even when other variables were considered. The receiver operating characteristic (ROC) curve analysis showed a cutoff value of HbA1c at 6.05% (42.6 mmol/mol) (area under the curve [AUC] = 0.615, *p* = 0.057) and of ACI at 2.08% (AUC = 0.650, *p* = 0.013), respectively.

### 3.3. Interaction Analyses

Figure 2 shows the interaction analyses of aOR for HbA1c and ACI. There were no interactions between male and female groups (HbA1c, *p* = 0.640; ACI, *p* = 0.520; *p*-value for interaction). The impact of HbA1c and ACI tended to be observed more in male donors. 

### 3.4. Distribution of Preoperative Predictors by Pathological Scores

Figure 3 demonstrates the relationships between preoperative factors and pathological scores. HbA1c showed a significant correlation with ct scores (*p* = 0.035; *p*-values for Kruskal Wallis test). ACI also showed a positive correlation with ci/ct/ah scores, respectively (ACI and ci, *p* = 0.005; ACI and ct, *p* = 0.021; ACI and ah, *p* = 0.017; *p*-values for Kruskal Wallis test).

## 4. Discussion

The present study revealed that HbA1c and ACI could be noninvasive preoperative indicators for the prediction of pre-existing histopathological damages in baseline kidney allografts. While HbA1c did not differ significantly between the two groups, the potential impact of HbA1c was uncovered after multivariable analyses. We hypothesize that no significant difference was found because the number in the CC group was relatively small and was affected by other factors included in the multivariate analysis.

We have previously shown that pre-existing histopathological damages (ci + ct ≧ 1 ∩ ah ≧ 1) were correlated with an insufficient one-year recovery of the residual renal function [7]. However, factoring baseline biopsy results into selecting living donors is impractical because it is currently impossible to estimate the ”quality” of a donated kidney without biopsies. In the present study, we could elucidate the possible preoperative predictors that reflect the histopathological state of donating kidneys.

The present study indicated that living donors with moderately high HbA1c may have a positive ct score. Glucose intolerance may become worse after unilateral nephrectomy [34], and microvascular injury and microalbuminuria may be present in prediabetes as an impaired glucose tolerance [35]. A prior study reported that individuals with glucose intolerance and without diabetic complications did not develop ESRD after donation [36]. However, they did not review pathological findings at the time of the donation. Considering the pathological perspective, IF/TA and interstitial inflammation were independent variables associated with renal prognosis in diabetes mellitus type 2 patients [37]. Mise et al. also reported that the progression of glomerular tubulointerstitial and vascular lesions were associated with renal death in patients with diabetic nephropathy [38]. These pathological findings have been recognized in the latter stage of developing diabetes mellitus. The present study added the fact that moderately high HbA1c may be associated with tubular atrophy before the presence of diabetic symptoms. In fact, one of the biggest causes of declination to living donations is the presence of impaired glucose tolerance [39]. Early intervention programs for candidates could increase the number of persons eligible for kidney donations.

Aortic calcification also tended to be associated with pre-existing histopathological damages of allografts in the present study. Consistent with our findings, a previous study reported that living donors with aortic calcification had a higher than average probability of delayed renal function recovery after donation [40]. Aortic stiffness of living donors may also contribute to the recipient outcome beyond other parameters [41]. Another report also demonstrated that allografts from donors with vascular calcification had a high percentage of vascular fibrous intimal thickening and arteriolar hyaline thickening in the six-month surveillance biopsy [42]. The present study revealed that a higher ACI was associated with higher ci, ct, and ah scores. The present study was in line with prior studies and added that even a low ACI may be associated with arteriolar hyalinosis as well as IF/TA in the baseline allografts. While ACI is not a perfect alternative for arteriosclerosis and aortic stiffness, ACI may be one of many noninvasive markers to identify the degree of arteriolar hyalinosis in donated kidneys.

The degree of sarcopenia did not show a correlation with pre-existing histopathological damages in the present study. Sarcopenia was highly prevalent in elderly patients with ESRD (approximately 30%) [43]. Although there are many studies regarding the impact of sarcopenia on graft outcomes [44,45,46,47], the relationship between a donor’s sarcopenia and renal function is little known. Further studies are needed to investigate the relationship between preoperative sarcopenia and renal function.

There are gender differences in the occurrence of hypertensive kidney disease. Though we could not prove the relationship between female preoperative biomarkers and subclinical kidney dysfunction, the ah score was significantly associated with a history of hypertension, especially in women. Female kidneys were more vulnerable to hypertension [48], and the rate of renal disease progression was faster in women with hypertension than in men with hypertension [49]. The Italy Developing Education and Awareness on Microalbuminuria in Patients with Hypertensive Disease study reported that women had lower GFR than men despite comparable blood pressures [50]. Consistent with former studies, the present study reinforced that female kidneys were vulnerable to a hypertensive state.

Subclinical kidney damages were not proven to be associated with the donor age in the present study. In Japan, living donors were seen to be getting older as there are many kidney transplantations from spouses and parents due to the shortage of organs [1]. The decreasing rate of kidney function after donation in elderly donors was almost equal to young donors [51,52]. Therefore, donation from elderly people tends to be acceptable. However, the present study suggested that subclinical statuses such as HbA1c and ACI reflected “biological” aging in living kidney donors. We need to start a larger study on the relationship between preoperative factors and biopsy findings, especially among elderly donors.

### Limitations

Some limitations of the study should be acknowledged. First, this was a single-center cross-sectional study, and the sample size was relatively small; there were only 27 donors in the CC group. Second, the possibility of intraoperative influences was not considered when interpreting 1-h biopsies. Some troubles during surgery, such as bleeding from the anastomosis and acute tubular necrosis after reperfusion, may affect the kidney biopsy result [26]. These influences should have been accounted for. Third, a type of selection bias may exist in this study. All donors may hope to help their families or their spouses by donation. Therefore, participants may be particularly careful of their own health so as to be eligible to be donors. Fourth, the population of the present study was homogenous and made up of a single ethnicity. It would be hard to adapt Japanese data to other ethnicities.

While HbA1c and ACI tended to be associated with pre-existing histopathological damages as mentioned above, their impact may be relatively limited. There are two reasons. First, the ROC curve analysis showed a relatively low AUC (HbA1c, 0.615; ACI, 0.650). This indicated that the accuracy of the cutoff values may be not very reliable. Second, in Figure 3, the correlation between HbA1c and the chronicity subscores of the Banff classification was proven only in relation to the ct scores, while the correlation between ACI and the chronicity subscores could be shown. On the basis of histopathological findings of diabetic nephropathy [37], these results may have been contributed to by the small sample size of donors. The present study could state that HbA1c and ACI were independent risk factors for predicting pre-existing histopathological damages; however, further research may be needed to reveal more reliable results regarding HbA1c and ACI.

## 5. Conclusions

This study suggested that HbA1c and ACI might be independent risk factors of pre-existing histopathological damages in the allograft. These noninvasive markers should be factored into the living donor selection process as well as into the careful follow-up of living donors.

## Figures and Tables

**Figure 1 jcm-09-03266-f001:**
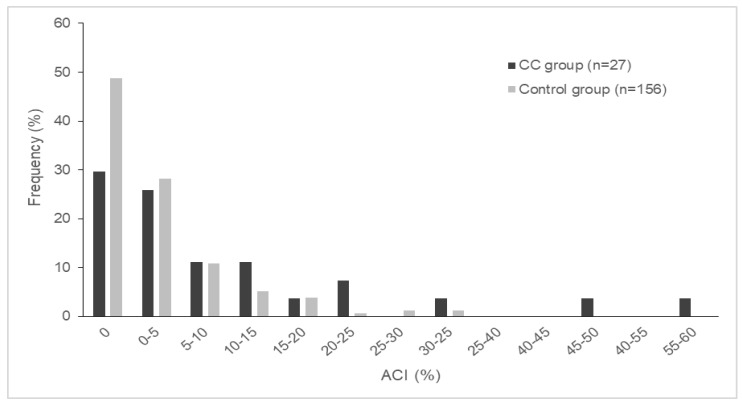
The distribution of ACI. ACI: abdominal calcification index. CC: chronic change.

**Figure 2 jcm-09-03266-f002:**
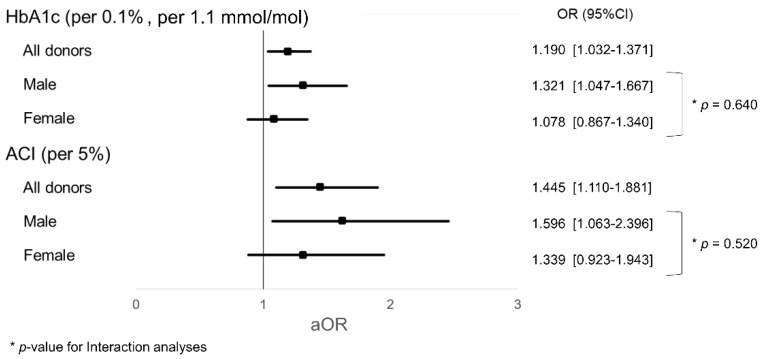
Interaction analyses for HbA1c and ACI. HbA1c hemoglobin A1c. ACI: abdominal calcification index. aOR: adjusted odds ratio.

**Figure 3 jcm-09-03266-f003:**
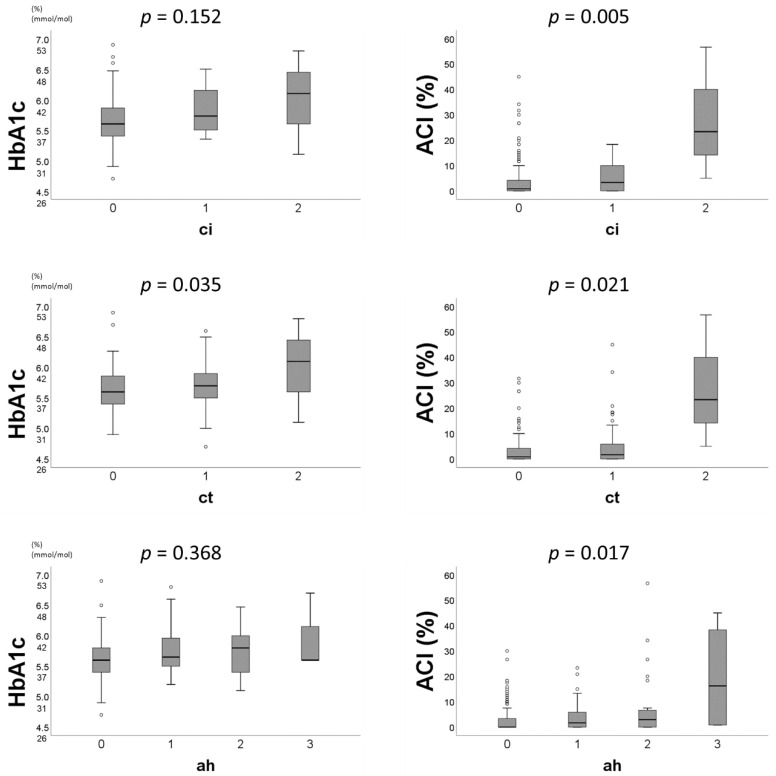
The distribution of HbA1c and ACI by ci, ct, and ah score. HbA1c: hemoglobin A1c. ACI: abdominal calcification index. ci: interstitial fibrosis. ct: tubular atrophy. ah: arteriolar hyalinosis.

**Table 1 jcm-09-03266-t001:** Baseline characteristics.

	All Donors (*n* = 183)	Chronic Change (CC) (*n* = 27)	Control (*n* = 156)	*p* Value
Sex: male (%)	65 (35.3)	13 (48.1)	52 (33.3)	0.138
Age (years)	58 (51.0-65.0)	60 (52.0–69.0)	58 (50.0–65.0)	0.174
Tobacco use, *n* (%)				0.035
Nonsmoker	102 (55.4)	10 (37.0)	92 (59.0)	
Current smoker	32 (17.4)	9 (33.3)	23 (14.7)	
Ex-smoker	18 (9.8)	3 (11.1)	14 (9.0)	
BP (mmHg)	124 (115.0–136.0)	124 (118.0–133.0)	123 (114.3–136.8)	0.381
Hypertension, *n* (%)	59 (32.1)	11 (40.7)	48 (30.8)	0.306
HbA1c (%)	5.6 (5.4–5.9)	5.7 (5.5–6.1)	5.6 (5.4–5.8)	0.057
(mmol/mol)	38.0 (36.0–41.0)	39.0 (37.0–43.0)	38.0 (36.0–40.0)	
Uric acid (mg/dl)	4.9 (4.2–5.7)	5.4 (4.6–6.3)	4.8 (4.1–5.6)	0.029
LDL (mg/dl)	120 (102.0–138.0)	123. (101.0–137.0)	118 (103.3–138.8)	0.766
ACI (%)	0.83 (0.0–5.0)	3.33 (0.0–13.3)	0.83 (0.0–4.2)	0.009
BMI (kg/m^2^)	22.9 (20.9–25.1)	22.7 (21.5–25.4)	22.9 (20.8–24.9)	0.473
PMI (cm^2^/m^2^)	4.40 (3.69–5.57)	4.68 (3.72–5.68)	4.36 (3.69–5.56)	0.511
eGFR (ml/min/1.73m^2^)	81.1 (73.0–91.8)	86.2 (74.0–94.7)	81.1 (73.0–91.6)	0.386
ci, *n* (%)				<0.001
0	160 (87.4)	18 (66.7)	142 (91.0)	
1	20 (10.9)	6 (22.2)	14 (9.0)	
2	3 (1.6)	3 (11.1)	0 (0)	
3	0 (0)	0 (0)	0 (0)	
ct, *n* (%)				<0.001
0	114 (62.3)	1 (3.7)	113 (72.4)	
1	66 (36.1)	23 (85.2)	43 (27.6)	
2	3 (1.6)	3 (11.1)	0 (0)	
3	0 (0)	0(0)	0 (0)	
ci + ct ≥ 1, *n* (%)	71 (38.8)	27 (100)	44 (28.2)	<0.001
ah, *n* (%)				<0.001
0	123 (67.2)	0 (0)	123 (78.8)	
1	30 (16.4)	13 (48.1)	17 (10.9)	
2	26 (14.2)	13 (48.1)	13 (8.3)	
3	4 (2.2)	1 (3.7)	3 (1.9)	

Median (IQR). BP: blood pressure. HbA1c: hemoglobin A1c. LDL: low-density lipoprotein. ACI: abdominal calcification index. BMI: body mass index. PMI: 3rd lumber psoas muscle index. eGFR: estimated glomerular filtration. ci: interstitial fibrosis. ct: tubular atrophy. ah: arteriolar hyalinosis.

**Table 2 jcm-09-03266-t002:** Independent predictors associated with pre-existing histopathological damages.

	Univariable Analysis		Multivariable Analysis	
	OR (95%CI)	*p* Value	OR (95%CI)	*p* Value
Sex (ref. female)	0.538 (0.236–1.229)	0.141		
Age (years, per 1)	1.030 (0.987–1.076)	0.176		
Tobacco use	1.704 (0.945–3.073)	0.076	1.485 (0.769–2.868)	0.239
BP (mmHg, per 1)	1.013 (0.987–1.039)	0.343		
Hypertension	1.547 (0.668–3.582)	0.309		
HbA1c (%, per 0.1) (mmol/mol, per 1.1)	1.144 (1.023–1.278)	0.018	1.190 (1.032–1.371)	0.016
Uric acid (mg/dl, per 1)	1.369 (1.011–1.853)	0.042	1.087 (0.747–1.583)	0.663
LDL (mg/dl, per 1)	1.001 (0.988–1.015)	0.843		
ACI (%, per 5)	1.413 (1.132–1.763)	0.002	1.445 (1.110–1.881)	0.006
BMI (kg/m^2^, per 1)	1.050 (0.918–1.201)	0.473		
PMI (cm^2^/m^2^, per 1)	1.032 (0.806–1.323)	0.801		
eGFR (ml/min/1.73m^2^, per 1)	1.010 (0.983–1.037)	0.483		

BP: blood pressure. HbA1c: hemoglobin A1c. LDL: low-density lipoprotein. ACI: abdominal calcification index. BMI: body mass index. PMI: 3rd lumber psoas muscle index. eGFR: estimated glomerular filtration. CI: confidence interval.

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
