# Peer review of "HbA1c and Aortic Calcification Index as Noninvasive Predictors of Pre-Existing Histopathological Damages in Living Donor Kidney Transplantation"

_jcm, 2020, doi:10.3390/jcm9103266_

Round 1
Reviewer 1 Report
Authors need to elaborate on clinical significance of their study:
1. Donor contribution to recipient outcome goes beyond simple parameters like age, gender and even familial or non-familial donor type. REF: Bahous SA, Stephan A, Blacher J, Safar M (2012) Cardiovascular and renal outcome in recipients of kidney grafts from living donors: role of aortic stiffness. Nephrol Dial Transplant 27(5):2095–2100
2. Functionally does aortic calcification have consequences in terms of pulse pressure, a surrogate of aortic stiffness in this relatively healthy donor population?
3. Donation of a kidney modulates glucose metabolism. Unilateral nephrectomy has been shown to impair insulin sensitivity [Shehab-Eldin W, Shoeb S, Khamis S et al: Susceptibility to insulin resistance after kidney donation: a pilot observational study. Am J Nephrol, 2009; 30: 371–76. microvascular injury and microalbuminuria may be present in prediabetes as a consequence of altered glucose metabolism . [Singleton JR, Smith AG, Russell JW, Feldman EL: Microvascular complications of impaired glucose tolerance. Diabetes, 2003; 52: 2867–73].
4. Since donors are motivated group from general population: Establishing Donor lifestyle intervention program can be encouraging and guide a prospective donor in lifestyle intervention trials, with an aim increase the number of persons eligible for kidney donation. [Guthoff M. et al.: Glucose metabolism in living kidney donors © Ann Transplant, 2016; 21: 39-45]
Reviewer 2 Report
In this cross-sectional study, dr. Tanaka et al. aim to compare donors with signs of biopsy-proven histopathological damage to donors without this damage. A study with such a large number of living donor renal biopsies should be able to answer important and relevant clinical questions and the authors have done a decent analysis on the subject. However, the manuscript should be substantially be improved before it is suitable for publication.
Major:
- While this study has great potential, it lacks focus and clarity. The risk factors investigated are not clearly defined and the rationale for the investigation is not properly given. Instead, the authors aim to create a vague and not scientifically founded profile of “fragility” and talk about “changes in histopathology” unfounded. This manuscript quality should be improved by defining the risk factors that can play a role in histopathological kidney damage and then explaining the found associations. The wealth of biopsy data are under-utilized by providing only associations without proper rationale, definition, and hypothesis.
- Since the donor selection criteria of these donors have an important impact on the results of the study, the authors should mention this in their methods section. The authors admit the study being vulnerable to selection bias but do not try to analyze the amount of bias.
- Discussion: the discussion connects poorly with the set goals of the introduction.
Other/minor:
- The authors state that the design of the study is a retrospective (cohort) study. However, the analysis that was performed is of cross-sectional nature. This should be changed in the manuscript.
- The authors use the term: “Chronic histopathological changes” and divide their donors into two groups: with or without changes. However, this terminology may not be the best. Whilst the study shows results of a large amount of biopsies, no serial biopsies have been performed. Therefore, no real statement on chronicity or ‘changes’ can be made. The authors should consider renaming “chronic histopathological changes” to “histopathological signs of pathology” or “damage”.
- Abstract (page 1): The authors should add a short sentence that explains that patients with a chronic change score of more than 1 fall into the group ‘chronic change’. Add the N of each group between brackets. This also should be added to the methods.
- Abstract (page 1): The authors should add the p-values of HbA1C and ACI in the abstract.
- Abstract (page 1): It is unclear what the authors exactly mean with subclinical predictors.
- Introduction (page 2): The authors state they investigate the impact of subclinical fragile status. They do not define this fragile status in their introduction and from the manuscript and data no clear label of “fragile” can be made. The authors may want to consider changing the terminology of “fragile status” to donors with risk factors (for kidney failure or disease). It is also advised to discuss which risk factors are tested and provide the rationale. The authors mention HbA1c and ACI in the title, but not once in the introduction. The authors should define a clear hypothesis and test it.
- Methods (page 3): The authors should explain what the subscores of the Banff classification mean (e.g. that ct refers to tubular atrophy and ah to arteriolar hyalinosis). They should also provide a rationale on why these domains of the Banff classification are used.
- Methods (page 3): The authors should provide more detail on their CT scan protocol.
- Methods (page 3): The authors state that data were expressed as median and IQR for continuous data. For normally distributed data, the mean (SD) would be a more appropriate representation of the data.
- Methods: The authors state: “The optimal cut-off points of preoperative continuous data in the prediction of chronic kidney damages were determined using ROC curve analysis, and the AUC was calculated to assess the accuracy.” They do not explain this analysis in the discussion and its added value is disputable.
- Results: The authors state in their title that HbA1c is associated with “chronic change”. However, the HbA1c does not significantly differ between the groups.
- Results: Multivariate analysis should be “multivariable analysis”.
- Results: ACI variation mentioned in the text is unclear in the table, maybe add additional figure to clarify whether patients with an ACI of 0% were part of the CC group?
- Results: What is the added value of figure 1 if the p-value is 0.520 and 0.640? Apparently there is a difference between males and females but not a significant difference. The authors incorrectly state that the role of HbA1c and ACI was more “significantly observed in male donors”.
- Results: The authors should discuss the figure in which they associated HbA1c and ACI with histopathological signs of damage. And then should provide an explanation in the discussion for why some parts are associated.
- Results: The authors may want to present HbA1c levels in accordance with the International Federation of Clinical Chemistry and Laboratory Medicine (IFCC) and report mmol/mol) as well.
- Discussion: the authors have investigated pre-implantation biopsies. This means that signs of damage reflect the “kidney quality” of the donor. This is the most important implication of the study: HbA1c and ACI may be non-invasive markers of living kidney donor “quality”. The authors should explain this further.
- Discussion: it is unclear why the authors discuss PMI and sarcopenia. PMI is not associated with the primary outcome and is not likely to be very prevalent in Japanese kidney donors accepted for donation. I suggest to remove this section form the manuscript or reduce it to 1 sentence.
- Discussion: The authors state ” In this study, we newly establish the standard value of PMI in elderly Japanese population.” This is a false statement. The study was not design nor reported to establish this. Either the authors should add this to their methods, results, and then provide a better analysis of reference values (e.g. 95pct analysis) or remove this part.
- Discussion: The authors state “The personal difference between biological aging and actual aging of living donors may be highest in their 50-60s”. They should add an explanation of this statement or a reference to evidence for this statement.
- Discussion: The generalizability of the Japanese data for the worldwide donor population should be discussed.
- Conclusions: The authors conclude that prediabetes and early stages of arteriosclerosis are associated with histopathological signs of damage. However, they did not define prediabetes of arteriosclerosis. To draw this conclusion, the authors should provide a more detailed analysis and definition.
- The authors should revise the manuscript for use of the English language. Some examples:
- Introduction (page 2): “impact of chronicity score” should be “impact of the chronicity score”.
- Chronic change can be changed to the abbreviation CC. This abbreviation is used earlier in the manuscript in the methodology but should maybe be explained earlier in the introduction.
- Discussion (page 7): Sentence 1. Delete ‘the’ noninvasive preoperative…..
- Discussion (page 8): ", it is no wonder that elderly people tend to have biological aging." This statement should be revised for scientific use of language.
- Conclusions (page 9): Prediabetic should be prediabetes.
Reviewer 3 Report
In this single center, retrospective cohort study, the authors asked whether clinical parameters of microvascular injuries are associated with chronic histopathological changes in living donors. Authors collected imaging and clinical parameters as well as 1h-post-implantation biopsy of 187 consecutive living donors at their institution. Key finding is that HgbA1c and ACI are both independent risk factors associated with histopathological chronic changes in kidney biopsy.
Overall, the study is well conducted, and the conclusion is reasonably supported by the data. The study is meaningful because it analyzed a clinical and histopathological correlations in kidney donors, the population who usually did not get biopsied otherwise. It is noteworthy that histopathological changes are already present even in donors with apparently “normal” A1c or other clinical signs of microvascular disease.
Major comments:
It would be clinically more meaningful if the authors could analyze the correlation between the clinical parameters of microvascular changes (HgbA1c and ACI etc.) and the donor’s clinical trajectory after donation. Did the chronic change group donors did have worse clinical outcomes (e.g. 1, 3, 5 years post-transplant kidney function after donation, or newly diagnosed diabetes)? How about the recipient outcomes from these donors?
Minor comments:
On page 2, please make coefficient (-1.094) and (-0.28) superscript. The reviewer understands that MDRD equation overestimates GFR in Japanese population, which could justify the use of Japanese society of nephrology calculation. Please add reference(s) for this calculation so the readers worldwide could understand the approach.
On page 3, please clarify the definition of ACI. Is there signal cut off (e.g. Hounsfield unit) to be considered as positive calcification?
Round 2
Reviewer 2 Report
The authors have substantially improved the manuscript. I have a few more minor suggestions to further improve the manuscript.
- Abstract: add to the first sentence that the scorings used are measures of the Banff classification.
- Introduction: it may be beneficial to the readability of the abstract to add a reference/and or explanation of the chronicity score when the authors first use it.
- Introduction: I suggest explaining a bit more the pathophysiological cause of deterioration of renal function because of metabolic syndrome: the manuscript depth could be further increased by discussing hyperfiltration in donation/transplantation (and provide references to relevant literature).
- Methods: “The definitions of ACI and PMI were described afterward.”. The authors should re-write this sentence for clarity.
- Figure 1 has is a nice addition to the manuscript, but should be discussed in the results or should be discussed below the figure.
- Discussion: The authors state: “The reason why there were no significant difference was that the number in the CC group was relatively small and was affected by other factors included in multivariate analyses.”. This is too briskly put. The authors could re-write this sentence to: “We hypothesize that no significant difference was found because…. “
- Discussion: ‘’The present study…diagnosis of diabetes mellitus.’’ It is not proven in the present study that HbA1c affects interstitial fibrosis and arteriolar hyalinosis. It is shown that HbA1c is associated with tubular atrophy (ct, p=0.035). I suggest re-writing this sentence to reflect the results.
